behaviour/cognition

adoption, alloparental care, parental care, cooperation, vampire bats

**Author for correspondence:**
Imran Razik
e-mail: razik.2@osu.edu

# Non-kin adoption in the common vampire bat

Imran Razik[1,2], Bridget K. G. Brown[1,2], Rachel A. Page[2] and Gerald G. Carter[1,2]

[1]Department of Evolution, Ecology, and Organismal Biology, The Ohio State University, 318 W. 12th Ave, Columbus, OH, USA
[2]Smithsonian Tropical Research Institute, Balboa, Ancón, Republic of Panamá

 IR, 0000-0002-8529-6212; BKGB, 0000-0001-9774-3887;
RAP, 0000-0001-7072-0669; GGC, 0000-0001-6933-5501

Individual animals across many different species occasionally 'adopt' unrelated, orphaned offspring. Although adoption may be best explained as a by-product of adaptive traits that enhance parental care or promote the development of parental skills, one factor that is possibly important for the likelihood of adoption is the history of cooperative interactions between the mother, adopted offspring and adopter. Using 652 h of behavioural samples collected over four months, we describe patterns of allogrooming and food sharing before and after an instance of non-kin adoption between two adult female common vampire bats (*Desmodus rotundus*) that were captured from distant sites (340 km apart) and introduced to one another in captivity. The first female died from an illness 19 days after giving birth. The second female groomed and regurgitated food to the mother more often than any other group member, then groomed, nursed and regurgitated food to the orphaned, female pup. The substantial increase in alloparental care by this female after the mother's death was not observed among the 20 other adult females that were present in the colony. Our findings corroborate previous reports of non-kin adoption in common vampire bats and are consistent with the hypothesis that non-kin adoption can be motivated, in part, by a history of cooperative interactions.

## 1. Introduction

Animals are most likely to care for others' young (i.e. alloparental care) in groups where members are related and helpers can gain indirect fitness benefits [1,2], as in most cooperative breeding societies [3,4]. In some cases, alloparental care towards either kin or non-kin could also result in direct fitness benefits for the helper (e.g. group augmentation [5,6]). However, alloparental care can also arise when non-adaptive, misdirected parental care [7–9] is triggered by a combination of rare circumstances, low costs of helping and/or imperfect kin discrimination [1,10].

One extreme form of alloparental care is the adoption of orphaned young. Adoption has been reported across 120 mammal species, including marsupials, shrews, primates, rodents, canines, ungulates, elephants, hyraxes, cetaceans and bats [1]. In some cases, individuals adopt kin, through which they might gain indirect fitness benefits [1,11]. In other cases, individuals adopt familiar non-kin (e.g. chimpanzees: [12]; cercopithecines: [13]; muriquis: [14]; phocid seals: [15]), or unfamiliar or seemingly unfamiliar non-kin (e.g. dolphins: [16]; Barbary macaques: [17]; Rhesus macaques: [18]). In humans, the adoption of kin or non-kin children is well described in both traditional and modern industrial societies around the world [19]. Whether or not adopting non-kin is adaptive remains unclear.

Non-kin adoption could be explained as a non-adaptive by-product of adaptive traits that enhance parental care or promote the development of parental skills. For instance, adoption might arise as a by-product of traits that evolved to increase interest in conspecific young [1]. In several primate species, females often appear interested in handling others' infants and may even try to kidnap infants from other group members, which may be associated with reproductive seasonality or adolescence [13,20,21]. Given that decision-making is also imperfect and constrained, parents may face a trade-off between the cost of mistakenly providing occasional care to non-offspring and the cost of failing to respond to signals of need from true offspring [1,22]. Nevertheless, although adoption may not be adaptive, it can be inherently rewarding to the adopter at the proximate level, as in the human motivation to adopt unrelated children [19]. The cognitive and neuroendocrine mechanisms that underlie parental care are an inherently interesting field of study, and detailed accounts of adoption in different species can shed light on the shared proximate factors that motivate the decisions of individuals to adopt non-kin young.

One possible, yet understudied, factor influencing non-kin adoption is the history of cooperative interactions between the mother, offspring and adopter. In non-human primates, some evidence suggests that pre-existing relationships play an important role in an individual's decision to adopt an orphaned infant [11–13]. If so, our understanding of non-kin adoption could be improved by tracking cooperative interactions across different relationships through time.

We tracked the allogrooming and food-sharing relationships among 23 adult female common vampire bats (*Desmodus rotundus*) before and after one female adopted the orphaned pup of another female bat that was unrelated and previously unfamiliar. The two adult females, named 'BD' and 'Lilith', first met in captivity after being captured from sites 340 km apart. Lilith, the bat that gave birth and died, was near the end of its pregnancy at the start of the study. BD, the bat that became the 'adopter', was previously housed in captivity for another study almost 2 years prior (see materials and methods). We observed no evidence that BD was either pregnant or lactating when first introduced to Lilith, nor did she show any signs of pregnancy for the next four months. Our findings corroborate previous reports of adoption in captive vampire bats [23,24] and provide evidence that adoption may be motivated, in part, by a history of cooperative interactions.

# 2. Material and methods

As part of an ongoing study on the formation of cooperative relationships, we combined three wild-caught groups of common vampire bats into a single, captive colony at the Smithsonian Tropical Research Institute in Gamboa, Panamá. We primarily captured female bats that appeared fully developed [25,26], which were classified as 'adults', while younger bats were classified as a 'juvenile', if alone, or 'pup', if attached to the mother. From a cave at Lake Bayano, Panamá, we captured nine bats, including six adult females, two juvenile males and one juvenile female. From outside a large, hollow tree roost in Tolé, Panamá, we captured seven adult females. From outside another hollow tree roost in La Chorrera, Panamá, we captured eight adult females. These sites were 120–340 km apart and were chosen to ensure that the bats from each site were unrelated and unfamiliar.

To identify bats, we marked individuals with a unique set of forearm bands (Porzana and National Tag). On 14 June 2019, groups were simultaneously released into an outdoor flight cage (2.1 × 1.7 × 2.3 m). On 5 July 2019, we added two more adult female bats from the capture site in Tolé, Panamá. These two bats, BD and BSCS, were banded re-captures from a past study; they were previously captured from the wild roost, studied in captivity from December 2015 to September 2017, then released back into the wild roost (for details see [27,28]).

To measure dyadic social interaction rates, we used three infrared surveillance cameras (Foscam NVR Security System) to sample allogrooming and food-sharing interactions among the captive bats for 6 h each day from 23 June to 4 August 2019 and from 11 August to 14 October 2019. We sampled in 1 h

periods, primarily at 0400, 0500, 0900, 1900, 2000 and 2100 h. On 3 of 110 days, we sampled only 3–4 h, and for 18 days, we sampled at other hours in between. During 640 sampling periods, we recorded all dyadic bouts of allogrooming and mouth-licking (possible food sharing) that were at least 5 s in duration, noting the actor and receiver. For allogrooming, we identified the actor as the individual licking the body of the receiver. For possible food-sharing bouts, the actor was identified as the individual regurgitating food to the receiver, which we inferred from mouth-licking by the recipient while the actor held still (following [28,29]). To induce food sharing for specific individuals, we individually isolated up to three bats each day and fasted them for 23 h. Each bat was fasted 3–15 times, and bats were mostly fasted once every seven days. Bats were weighed before and after fasting. Each night, for 8–12 h, all other bats were provided with cattle blood that was defibrinated with 44 g of sodium citrate and 16 g of citric acid per 19 l container. Cattle blood was either refrigerated for short-term use or stored frozen, then thawed for later use.

Our analyses of adoption behaviour were motivated by the following events. Four females gave birth while we were collecting behavioural data. One of these females, Lilith, gave birth on 9 August 2019 while being monitored within a separate observation cage. To sample interactions between Lilith and the newborn pup, we used an infrared camera to record them for an additional 12 one-hour periods between 9 and 11 August 2019, resulting in a total of 652 sampling periods. We then returned the mother and pup to the larger group. Three weeks after giving birth, Lilith died due to what we suspect was a gastrointestinal illness. Lilith's orphaned pup then received extensive allomaternal care from another adult female in the colony, BD, and survived until the end of the experiment on 14 October 2019, a span of approximately seven weeks (figure 1a). Allomaternal care from BD included grooming, regurgitated food sharing and nursing, which we confirmed by manually expressing milk from the nipple on the day Lilith died.

To calculate hourly rates of dyadic allogrooming and food sharing before and after the mother's death, we summed the total duration of dyadic interaction bouts for each sample hour during which each dyad had the opportunity to interact. To visualize changes through time, we used local polynomial regression fitting (loess method in the R package ggplot2 with a span argument of 0.8) to plot allogrooming and food-sharing rates between (i) the mother and its pup, (ii) all other adult females and the pup and (iii) the mother and all other adult females. To see if the increase in BD's helping behaviour was greater than expected by chance, we measured the mean change in allogrooming and food-sharing rates towards the pup before versus after the mother's death for all 21 adult females that were present in the colony during both periods (one female died before the mother gave birth). We then calculated the exact probabilities that BD would have the highest increases by chance (non-parametric test) and we used Grubb's test to detect if BD's increase in allogrooming and food-sharing towards the pup was an outlier (parametric test). For Grubb's test, we used log(seconds + 1) to increase normality of the allogrooming and food-sharing rates before calculating the mean change. We conducted all analyses in R version 3.6.1 [30].

## 3. Results

Two unfamiliar and unrelated adult female common vampire bats formed a new allogrooming and food-sharing relationship. Lilith, the bat that eventually gave birth, first entered the captive colony on 14 June 2019. BD, the bat that later adopted Lilith's orphaned pup, first met Lilith on 5 July 2019. At this time, BD was not noticeably lactating. Directed allogrooming rates between BD and Lilith were similar, highly symmetrical, and increased over approximately eight weeks, after which allogrooming rates were relatively stable (figure 1b). BD became Lilith's top allogrooming partner, while most other bats did not groom Lilith much or at all (figure 1b). BD was also Lilith's top food donor. Based on mouth-licking times, we estimate that 47% of Lilith's total mouth-licking (begging or receiving food) was directed to BD. Moreover, BD shared food with Lilith during two trials where Lilith was fasted, as inferred from mouth-licking and a subsequent increase in Lilith's mass.

On 9 August 2019, Lilith gave birth to a female pup (figure 1, vertical dashed line). In the following weeks, BD extended its cooperative behaviour from Lilith to Lilith's pup. As Lilith groomed and shared food with its pup less through time, BD's rates of allogrooming and food sharing with Lilith's pup increased (figure 1). During this time, observations from video footage suggested that BD began to nurse the pup, which seemed to increase gradually. By contrast, other colony members rarely groomed or shared food with Lilith's pup (figures 1 and 2).

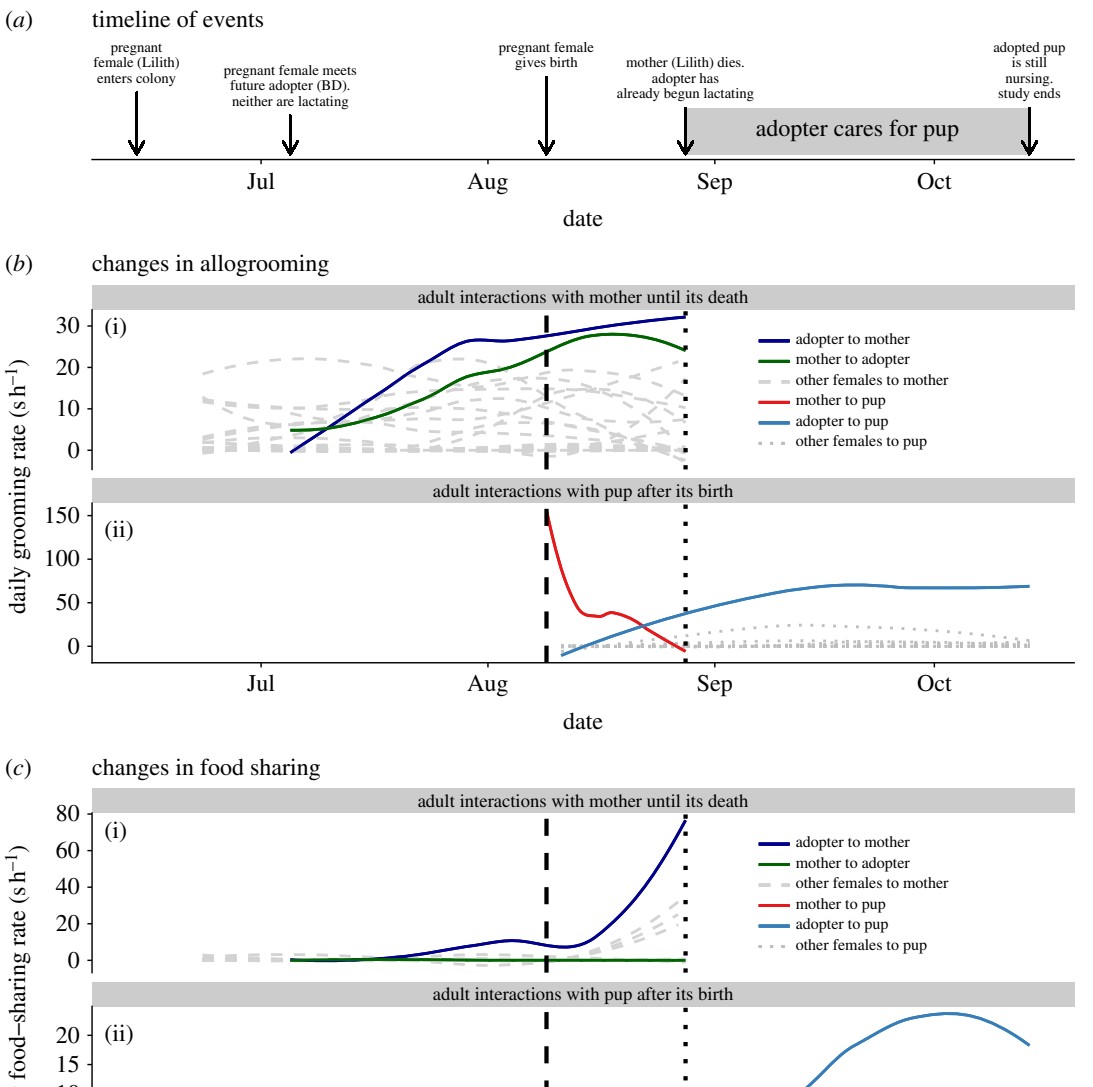

**Figure 1.** Evidence of adoption. (*a*) Notable events relating to the adoption are plotted chronologically from when the mother entered the captive colony to when the study ended. When first introduced to the captive colony, the mother was pregnant, and the female that later adopted the orphaned pup was not lactating. (*b*) The mother and the adopting female reciprocally and increasingly groomed each other for approximately eight weeks (dark blue and green lines), while 21 other adult females groomed the mother less (grey dashed lines) or not at all (not plotted). The vertical dashed line indicates when the female pup was born, while the vertical dotted line indicates when the mother died. After the mother gave birth, it groomed its pup less through time until it died due to a suspected gastrointestinal illness. The 'adopter' increasingly groomed the pup following the death of its mother. (*c*) The adopter increased its rate of food donations to the mother prior to the mother's death and became the mother's top food donor. The adopter also became the top food donor to the orphaned pup.

After giving birth, Lilith fell ill. During the week of Lilith's death, we observed that Lilith spent 96 min mouth-licking seven adult females, including BD. This amount of begging is a dramatic increase from the week prior, during which time Lilith spent just 2 min mouth-licking two adult females, again including BD. Lilith was the potential donor for only 13 min of recorded mouth-licking interactions with all other adult females while she was alive (figure 1*c*).

On 28 August 2019, Lilith died due to what we suspect was a gastrointestinal illness (figure 1, vertical dotted line). On this day, we observed that BD was lactating and providing milk to the pup. After this day, BD increased its rate of allogrooming the pup by 47 seconds per hour (s h$^{-1}$) and food sharing

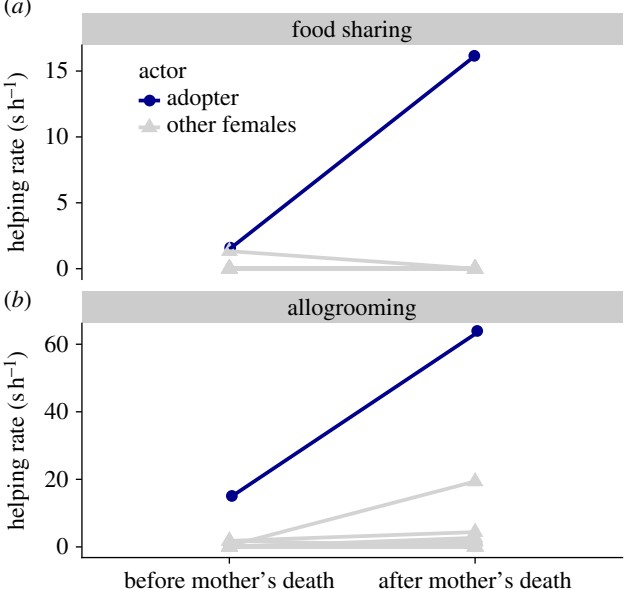

**Figure 2.** Increases in rates of allogrooming and food sharing from 21 adult females to the orphaned pup before and after the mother's death. The 'adopter' increased its allogrooming and food-sharing rates to the pup more than any other female. The allogrooming and food-sharing rates are shown for 21 bats, but most of the slopes are not visible because many of the rates are zero both before and after the mother's death.

with the pup by 15 s h$^{-1}$ (figure 2), which was greater than the 20 other bats (non-parametric rank test $p = 0.048$ for both allogrooming and food sharing; parametric outlier test, allogrooming: $G = 3.0$, $p = 0.007$, food sharing: $G = 4.0$, $p < 0.0001$). BD became the pup's highest ranked groomer and food donor, but BD was not the top groomer or food donor for any of the seven other juvenile bats in the colony. BD groomed the pup an average of 52 s h$^{-1}$ but groomed other juveniles only 0 to 4 s h$^{-1}$. BD fed the pup an average of 13 s h$^{-1}$ but fed other juveniles 0 to 1 s h$^{-1}$. BD was still nursing the pup when we finished observations on 14 October 2019.

Among all bats, BD was not exceptionally cooperative. Overall, BD ranked fifth for having the highest allogrooming and food-sharing rates towards other adult females. Besides Lilith and her pup, BD was also the top groomer for only one other bat (another bat from the same wild roost) and was not the top donor for any other bat. Moreover, BD's behaviour to other adult females changed after Lilith's death. Before Lilith's death, BD ranked third and eighth for highest allogrooming and food-sharing rates to other adult females (excluding Lilith). After Lilith's death, BD's rank decreased to ninth and 12th for highest allogrooming and food-sharing rates.

## 4. Discussion

We observed the complete social history leading up to a case of non-kin adoption between unrelated common vampire bats. In captivity, a mother gave birth then died after several weeks. During this time, another previously unfamiliar and unrelated female responded to changes in need of both the mother and the pup (figure 1). After the mother's death, the female increased the rate at which it allogroomed and regurgitated food to the orphaned pup. We also noticed that this female was providing the pup with milk. This increase in helping behaviour was uniquely directed to the pup (in comparison to other juveniles or adults in the colony), and no other female helped the orphaned pup to nearly the same extent (figure 2).

The probability of orphaned pups being adopted in vampire bats remains unknown. Uwe Schmidt [23] suspected that adoption was not an unusual behaviour in vampire bats based on an unreported number of cases where newly caught pups, whose wild-caught mothers died during transportation to the laboratory, were subsequently adopted by unfamiliar females in his captive colony. In each case, these adopting females began to lactate, as we observed here. In the same colony, captive-born pups that lost their mothers were also adopted by other females [24]. In one documented case, a pup of unreported sex lost its mother at 17 days of age, and an adult female then groomed, nursed and

shared food with this pup, but it did not survive. In another case, a juvenile male lost its mother at 79 days of age, and a female in the colony then began to share food with and nurse the orphaned male. This female was not lactating before her interactions with the orphaned pup; its previous pup was born 2 years prior and had died 16 months before the adoption. By tracking the weight of the juvenile male for 1 year, the authors found that the juvenile was dependent on the adopting female for milk and regurgitated food until an age of about 300 days [24]. Other group members, including males and unfamiliar females, occasionally shared food with the juvenile male [24]. Based on these observations, it was suggested that female vampire bats form stable social groups with communal care of young [24], but later studies in the field and laboratory provide evidence against communal offspring care, instead suggesting that cooperative behaviour occurs within individually differentiated relationships, rather than as group-level investments made indiscriminately to any group member [25,29,31–33].

Adoption is not commonly reported among other bats. In one documented case, a female lesser short-nosed fruit bat (*Cynopterus brachyotis*) adopted its orphaned grandson in captivity after losing a premature pup of nearly the same age [34]. Less extreme forms of alloparental care, such as allonursing and pup-guarding, are more frequently observed in some other bat species (reviewed by [35,36]).

It is important to note that the few observed cases of adoption in vampire bats all occurred in captivity. During more than 400 h of field observation, Wilkinson [25,31,32,37] observed allogrooming of vampire bat pups, but he never observed allonursing. Adoption would be difficult to observe in wild colonies because orphaned pups are uncommon. In our captive colony, it is also interesting to note that another female, BSCS, noticeably increased its allogrooming rate to the pup after the mother's death (figure 2). Both BD and BSCS had previously lived in captivity for almost 2 years during another study [27,28]. We therefore speculate that both captivity and death of the mother might increase the probability of adoption in vampire bats. However, the overall probability of adoption is difficult to estimate because observations of orphaned pups are rare.

There are several reasons why common vampire bats might present an interesting comparative case for future studies on the biological mechanisms behind adoption and maternal care behaviour. First, unlike most bats, female vampire bats often lack reproductive seasonality and can reproduce year-round, such that a female which adopts an unrelated pup will presumably reduce its own reproductive success, even if it recently lost a pup. Second, unlike most cooperative breeding mammals, an orphaned vampire bat pup is unlikely to be closely related to a random female in its colony because colonies have a low mean kinship, around $r = 0.08$ [31,38], and a single immigrating female bat would be likely to be unrelated to others encountered in the colony, which is the context simulated in our study. Third, relative to other bats, alloparental care poses an extreme energetic and opportunity cost. Female vampire bats give birth to a single pup after a gestation period of five to seven months [26,39], and a new mother will carry its pup for one to two months, during which time the weight of the pup will at least double. After four to five months, pups will have grown fourfold in mass and will begin to fly and feed on blood [26]. Weaning does not occur until approximately nine months of age [26,40]. By contrast, weaning in other bat species in the same family typically occurs after only one to three months [41–43]. Finally, reciprocal food-sharing behaviour among non-kin adults is likely to be based on the co-option of cooperative traits that originated for maternal care [28,30]. The same co-option of traits and expansion of cooperative behaviour from kin to non-kin may also help to explain non-kin adoption.

Our observation of non-kin adoption in vampire bats is similar in some respects to cases in chimpanzees [11,12]. Hobaiter *et al.* [11] suggested that chimpanzees may adopt an unrelated orphan if they previously experienced positive social interactions with that orphan prior to the mother's death. If so, then social interactions between adopter and adopted orphan may also be facilitated by a close social or affiliative relationship between the adopter and the biological parent. However, it is unknown if or how non-kin adoption based on previous social experience might affect lifetime fitness for both adopters and adoptees. Adoptive parents can experience energetic and opportunity costs from adoption that will vary across species, individuals and environments [1,13]. The survival of offspring would appear more likely after being adopted [1], but this is not always evident, especially among populations with high mortality [1,12,13].

Adoption can be explained completely, or in part, as a by-product of normal parental care. For example, if parental behaviours are triggered by a common set of neuroendocrine mechanisms or stimuli (e.g. [44]), then the same traits that lead to adaptive behaviour under typical circumstances could also cause non-adaptive adoptions under the rare circumstance of an orphaned infant in dire need. Moreover, the generally atypical circumstances associated with captive conditions may sufficiently increase the likelihood of non-kin adoption, which may not occur often or at all in the

wild. Some authors have suggested ways in which non-kin adoption might be adaptive for the adopter, including reciprocity [13,45], 'match-making' between biological and adopted young to form compatible mating pairs [46], or kinship deceit, by which adopters exploit kin-recognition heuristics and deceive adopted young into a false perception of kinship, thereby causing these young to later help at the nest [47]. Most, if not all, adaptive explanations predict that non-kin adoption of orphaned infants should be biased towards the philopatric sex, but this pattern has not been clearly shown [11–13]. In the case of vampire bats, it is also hard to explain why a female would invest in an asymmetric helping relationship with an infant over a potentially reciprocal relationship with an adult female [48], but these arguments do not eliminate the possibility that adopters may eventually receive social benefits from adopted individuals.

## 5. Conclusion

Observations of non-kin pup adoption in captive common vampire bats are consistent with the hypothesis that non-kin adoption is a by-product of proximate cognitive and neuroendocrine mechanisms that are crucial for parental care and triggered by unusual circumstances. Our observations also suggest that adoption is influenced by the history of cooperative interactions between mother, offspring and adopter, although more evidence is needed to test this hypothesis. The probability of non-kin adoption in vampire bats and whether it affects the fitness of adopters remains unclear, especially outside of captive conditions, but we can make two testable predictions. If non-kin adoption in vampire bats is adaptive, we predict that female pups should be adopted more often than male pups because females are philopatric. If female vampire bats are more motivated to help the offspring of more closely bonded partners, then vampire bat pups should be biased towards inheriting the cooperative relationships of their mothers, i.e. social inheritance of network ties [49].

Ethics. This work was approved by the Smithsonian Tropical Research Institute Animal Care and Use Committee (#2015-0501-2022) and the Panamanian Ministry of the Environment (#SEX/A-67-2019).

Data accessibility. Data and R code are available on Figshare (https://doi.org/10.6084/m9.figshare.13085129).

Authors' contributions. I.R. and G.G.C. conceived and designed the experiments. I.R., B.K.G.B. and G.G.C. performed the experiments and collected the data. I.R. and G.G.C. analysed the data and prepared figures. G.G.C. and R.A.P. coordinated the study and provided critical resources. I.R. and G.G.C. drafted the initial manuscript, and all authors revised it critically for intellectual content.

Competing interests. The authors declare that they have no competing interests.

Funding. I.R. was supported by a short-term fellowship from the Smithsonian Tropical Research Institute, a student research grant from the Animal Behavior Society and a graduate enrichment fellowship from The Ohio State University. B.K.G.B. was supported by a student research grant from Sigma Xi and a Critical Difference for Women Professional Development Grant from The Ohio State University.

Acknowledgments. We thank the Smithsonian Tropical Research Institute for providing logistical support and the following individuals who helped with vampire bat care and/or aspects of data collection: D. Aparicio, G. Cohen, L. Dück, D. Girbino, E. Kline, C. Marroquin, S. Ripperger and S. Stockmaier. We also thank M. J. West-Eberhard for initial conversations on this topic; J. K. Augustine, S. N. Gershman, I. M. Hamilton and the graduate students of The Ohio State University's EEOB Animal Behavior group for their useful suggestions during manuscript preparation, as well as the four anonymous reviewers for their insightful comments during the review process.

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
