## [Peer Review File · Royal Society Open Science]

Review History

RSOS-201927.R0 (Original submission)

Review form: Reviewer 1 (Óscar Chaves)

Is the manuscript scientifically sound in its present form?

Yes

Are the interpretations and conclusions justified by the results?

Yes

Is the language acceptable?

Yes

Do you have any ethical concerns with this paper?

No

Have you any concerns about statistical analyses in this paper?

Yes

Recommendation?

Accept with minor revision (please list in comments)

Comments to the Author(s)

In this short communication paper Razik investigated the non-kin adoption of an orphaned vampire bat (*Desmodus rotundus*) by an adult female (BD) after his mother (Lilith) death 19 days after giving birth in a captive bat colony in the SRI, Panama. Using three surveillance cameras, he collected 652 observation hours over four months to describe patterns of allogrooming and food sharing between the 23 adult females before and after the adoption event. After Lilith's died, the orphaned pup received allomaternal care from BD, and this infant survived until the end of the experiment. He concludes that the observation lend support to the hypothesis non-kin adoption can be motivated by a history of cooperative interactions.

This is a very well written paper on the proximal and evolutive causes of the of the allomaternal care and adoption of orphaned tropical bats. In the Introduction, Methods, and Results sections the author provided the necessary information to understand and study phenomenon (i.e. the role of inter-bat cooperative interactions on the allomaternal).

Certainly this work will be an interesting contribution on the topic. However, my main concern is that the extrapolation power of this study to the entire species is poor, because the lack of replicates (i.e. the number of allomaternal adoption events provided by the author is 1). Even, it is not clear for me if the main findings of this study could be extrapolated to free-ranging bats. Furthermore, most literature provide by the author in the discussion is based in captive bat colonies, then, allomaternal care may be a particular behavior of captive bats but not a 'natural behavior' of *Desmodus rotundus*. I understand that this type of events is, probably, very rare even in captivity, but the author must recognize the study limitations in the text and then, down the tone of the discussion.

Other issues

p.4, lines 7-24: Even when the author mention some appropriate statistical procedures to test the hypothesis, no statistical vales are provided in the Results section. This is a little strange and do not benefit the Result section, which result a largely descriptive section. Then, I author must indicate the statistical values of the tests in the text.

p.4, lines 32, 34, 35, 50, 52: The Figure 1 is not available in the proof provided by the editor. Only the Figure legend is provided even when this figure was important to understand the main findings.

p.5, lines 5-12: In this first discussion paragraph the authors repeat the main results. For me this paragraph must be removed. The discussion section must be focused in providing reasonable explanations to the main results, but not in synthetize the main results.

line 24: replace 'adoptive female' by 'adoptive mother'. The term 'adoptive female' has no biological sense. The same problem also is present in the Figure 1 according the R scripts provided.

p.5, lines 46-60: in this paragraph the author provided some reasons explaining why vampire bats are 'interesting comparative case' for studying the biological mechanism behind adoption. However, I suggest that this paragraph (or part of it) must be relocated to the introduction because this information do not contribute to explain the main results of this study.

p.6, lines 36-37: Because the reasons I have mentioned above, authors must down up the tone of this affirmation.

Review form: Reviewer 2

Is the manuscript scientifically sound in its present form?

Yes

Are the interpretations and conclusions justified by the results?

Yes

Is the language acceptable?

Yes

Do you have any ethical concerns with this paper?

No

Have you any concerns about statistical analyses in this paper?

No

Recommendation?

Accept with minor revision (please list in comments)

Comments to the Author(s)

This paper provides a detailed description of an interesting evolutionary problem – why female vampire bats adopt the offspring of unrelated females. A weakness of the study is that it details just one case of this unusual behaviour (which has been reported before) in a captive environment, but the behaviour is very well documented, the paper is clearly written and uses appropriate statistical analyses. I therefore believe it still makes a valuable contribution to the literature, and with a bonus that it was also enjoyable to read!

Major comments:

1. Could this adoptive behaviour just be an abnormal behaviour induced by captivity and not observed at all in the wild? This is discussed to some extent L30-38 on page 5 but I think it is worth emphasizing this as a potential caveat in later parts of the discussion covering how this adoptive behaviour may have evolved (particularly L16-29 of page 6).
2. I was surprised by the decision to permute both between time periods (by randomizing the direction of change) and between adoptive and non-adoptive females. My intuition is that this would mean a significant difference between the observed effect and the null effect could be driven either by adopter/non-adopter differences or by changes over time (e.g. all females provide more care in the period after maternal loss or just that food sharing and grooming increase over time as relationships strengthen between individuals that have spent more time in captivity together). By permuting both things we cannot tell which is driving this difference. It could be worth running this as two separate permutations, one asking does food sharing and grooming significantly increase between periods and another asking do they increase more between periods for the supposed adopter compared to other females?

Minor comments:

L27 p.2 and L15 p.13 – I was initially confused by what previously unfamiliar meant until I reached the methods. Is there a better way of phrasing this to making it clearer what is meant early on? E.g. previously unfamiliar females placed together within a captive colony.

L58 p.2 - Is it worth discussing this from the perspective of the individual being adopted as well as the individual doing the adopting? Could there be selection for traits in young bats that encourage alloparental care? I'm thinking about this from a parent-offspring conflict sort of perspective.

L55 p.4 - 'BD adopted lilith's pup' – it seems a bit too soon to say this. Might be better to first introduce the data and analyses that support this and then state it.

L58 p.4 – I think it needs to be made clearer that these 'expected increases' are 95% confidence intervals based on the null models. I ended up going through the code to double check.

Review form: Reviewer 3

Is the manuscript scientifically sound in its present form?

Yes

Are the interpretations and conclusions justified by the results?

Yes

Is the language acceptable?

Yes

Do you have any ethical concerns with this paper?

No

Have you any concerns about statistical analyses in this paper?

No

Recommendation?

Accept with minor revision (please list in comments)

Comments to the Author(s)

This interesting manuscript explores an exciting observation of adoption (including lactation in a previously non-lactating adult female) of a putatively unrelated pup. Adoption in non-human animals is poorly described in the literature especially amongst such well-studied mammals such as the vampire bats described here. To this end, I think this paper will be of broad interest. It is very well written and organized and the length seems appropriate. The biggest concern I have is how we can be certain that the 2 adult females are not related. Any evidence that could be provided to clarify this- even just greater explanations regarding the degree of relatedness normally observed at the same roost would be helpful. That said, some concerns may remain regarding how females disperse through the landscape and if low levels of relatedness at a single roost could be explained by females leaving at adulthood. If that were the case bats captured 100 km apart may be more likely to be related. Clarification in the text would at least help alleviate such concerns. The observations around lactation in a previously non-reproductive individual could be highlighted a bit more. Limitations of a single observed adoption (in this study) could be addressed in the discussion. Overall, I found this paper a fun and interesting read and I'm sure that others will feel similarly.

General comments.

Please explain how you can be certain that the two adult females are unrelated.

Please clarify (unless I am mistaken) that you do not know exactly when nor why BD was able to start lactating (re it was not truly 'spontaneous'). The older literature sometimes uses terms that have not withstood the test of time so I would see if there is a new and more appropriate way to describe this.

Would this observation of a single adoption have adaptive significance considering the Carter and Wilkinson labs raw number of hours watching vampire bats? It seems it must be extremely rare. Thus, it may be a fluke. That said it is fascinating and certainly thought-provoking. Nevertheless, you may wish to talk about this caveat as well.

Specific comments by lines unless otherwise noted.

Abstract.

22. Specify how it 'enhances parental care.' Do you mean they learn to be better parents?

23. Rephrase for clarity: "...adoption is the strength of previous cooperative..."

26. Specify that the sex of the pup is female.
 29. Explain what 'feeding' means here. Re. Blood sharing.

Introduction.

Page 2 of 8.

You only list indirect and nonadaptive benefits. Please also list 'adaptive' / 'non-indirect fitness' given the focus of your study.

44-51. I was surprised that humans were not discussed a little bit here. In an effort to indicate that evolutionary theory applies to humans and non-humans alike I would list them as one of your case examples here. I suspect some papers are available re- adoption amongst the Ache or Hazda.

54 - 55. Explain 'extreme attraction.' Maybe mention here that hormone levels may lead to displace parental caregiving.

Page 3 of 8.

5 - 9 Does this include humans (citations 9 - 11)?

19 - 21. Is this hypothesis that adoption is based in part on cooperative interaction something that has been discussed in bats before? If not, I would highlight here that this is the first time this idea has been postulated.

13, 14. Unrelated and previously unfamiliar. This idea is introduced here without any explanation. Fortunately, below you explain the distance between the sites (34,35). However, the reason they are believed to be unrelated should be explained earlier as it is key for the premise of your paper. That said, distance does not seem so large as to preclude them from being related. Towards the end of the manuscript, you mentioned the low r-value in vampire bats at the same roost. That information should also be given earlier (here is) to clarify that within a roost, vampire bats are not closely related. All that said, this is a key point that would ideally be addressed if you had any genetic material to verify the degree of relatedness between Lilith and BD. This also presents my only scientific concern with the manuscript and an important caveat to the study.

Methods general comments.

Where are Lilith and BD from (re - the 2 sites?)

53 - 56. Fasting is a (somewhat) unique state used experimentally in your study for obvious reasons. Do you think that this could have elicited the adoption behavior by inducing greater amounts of food sharing and thus adoption was a byproduct of greater interactions overall (maybe that increases 'priming' for adoption)? Could this explain why Wilkinson didn't observe adoption in naturally occurring free-living bats he studied?

Page 4 of 8.

3 - 6. I would reference your figure here.

Results.

References to figure one in the first paragraph made me think there was an illustration depicting how BD was not noticeably lactating. If you do not have any such pictures (which would be a nice addition if you did) I would move the reference to figure 1, so as to avoid this confusion.

31. Also in reference to the figure, I suggest adding important events such as Lilith meeting BD, Lilith being removed from the colony, and Lilith being returned to the colony. See my note in my reference to the figure below.

The first paragraph of results.

I was wondering if BD is not just a 'giver' perhaps with abnormally high oxytocin levels? Below you refer to the fact that she did not provide noticeably high amounts of care for other bats.

However, it is unclear when that observation was true relative to her adopting the pup. I think this needs clarification and reference to a key time point on your figure 1.

These second, third, and fourth paragraphs seem out of order. For example, you talk about Lilith giving birth in the third paragraph after you've talked about her decline after giving birth earlier on.

55 - 56. "On this day BD was found to be lactating." I would clarify in the text that she did not likely spontaneously start lactating exactly on the 28th of August. Unfortunately, you cannot rule out that she gradually started beforehand which, makes more sense physiologically. If you can say for certain, elaborate a bit on how you know or how you 'first noticed her lactating on...'. Discuss this caveat and how this relates to when we may have started supplementing the pup with milk. Please note on the next page line 24 through 25 you also mention spontaneously lactating. I would call this something else. For example, you could call it induced lactation or similar. Again, spontaneous would imply no hormonal nor behavioral priming which you have no evidence of (hormonal) and possible behavioral priming given earlier bouts of grooming.

Fourth paragraph line 60 to the end of page.

Here you refer to BD not being the top groomer nor food donor. However, it is unclear to me when this statement is for the whole study, before or after she adopts the pup. Please clarify.

Page 5 of 8 continued.

46 - 49. How often does a single female have more than one pup in a single year? If it is not particularly common, she may benefit more from taking care of a possible help her relative to trying to squeeze in yet another pup in a year. I'm curious which would be the more energetically beneficial strategy.

50 - 51. Here is where you discuss the low kinship value. Again, this ought to be mentioned much earlier as it is unclear how you know that these 2 females are unrelated.

Also, this paragraph.

I think you could highlight just how enormously expensive 9 months of lactation would truly be to help illustrate the cost of this unique behavior.

Page 6 of 8.

2 - 3. Could this occur vice versa where maternal care was co-opted to lead to the origins of food sharing? I suspected this could well be what happened in early humans.

Line 8.

Your discussion really focuses on the social interactions of the 2 adult females but here you discuss interactions with the orphan prior to the mother's death. Can you carefully parse these ideas up and either only focus on one or make sure you discuss the importance of interactions at both levels?

Paragraph lines 15 through 29.

I did not quite follow what was meant by "parental guided matchmaking, sexual imprinting" and to some extent even "deceit." If an adoptive mother is actually feeding a pup then she is truly giving the pup some benefit- so are you suggesting that the pup is expected to help later on but ultimately does not? Clarify.

Figure 1A timeline.

I would add important key events onto this figure. For example: When did the 2 adults meet? When was Lilith removed? When was Lilith returned? You could also add on hatched bars for the duration of lactation to highlight the period over which this is occurring for each of the 2 mothers.

For consistency sake I would use the names in parentheses on this figure as well.

Figures 1 B etc.

I would give a number in parenthesis dashed grey lines to highlight the focus possible number of interactions. This gets back to my previous question regarding whether BD isn't just a 'giver' and if her behavior changed after she adopted the pup.

Review form: Reviewer 4

Is the manuscript scientifically sound in its present form?

Yes

Are the interpretations and conclusions justified by the results?

Yes

Is the language acceptable?

Yes

Do you have any ethical concerns with this paper?

No

Have you any concerns about statistical analyses in this paper?

No

Recommendation?

Accept with minor revision (please list in comments)

Comments to the Author(s)

I have reviewed Rezik et al's manuscript titled 'Non-kin adoption in the common vampire bat'. I thought this was a well-written, succinct manuscript. The article describes a single incidence of offspring adoption in the common vampire bat. This behavior is of interest because it is exceptionally rare, especially among non-kin. Suspected adoption in common vampires was reported in a prior manuscript, but this is the first to make detailed observations that confirm the behavior.

I think the introduction was well written, touching on all appropriate background, and the methods were clear. The use of polynomial regression fitting was a reasonable approach to characterizing the data.

In the results, the timeline was a bit awkward. In the paragraph starting line 41, activities after Lilith's death were described. Then, the paragraph that starts on line 48 jumps back to describing when Lilith gave birth. To me, this would have better flow if the sentence starting 'On 9 August 2019..' was removed and the next started 'In the weeks following parturition,...'. The date of birth was described previously in the methods.

I thought the figures nicely portrayed the patterns described.

In the conclusion, the authors state that social interactions strongly impact non-kin adoption. While I agree that is likely, I strongly recommend that authors tone this down, as it is impossible to draw strong conclusions from a single adoption event.

Overall, I think this paper will be of interest to bat biology and the organismal biology community. The biggest problem with this manuscript is that it is based on a single observation of adoption. Thus, the authors and readers should not be confident that the observed behaviors

would be typical of adoption in this species, and it remains unclear how common this behavior is. In the past, journals such as the Journal of Mammalogy had a 'Notes' section for important natural history observations, such as this one. Few journals will publish this type of natural history observation based on a single observation – it's up to the editors if natural history observations have a home in the Royal Society Open Science. I think natural history observations are important, and it would be supportive of Open Science publishing this manuscript if it meets their standards.

Decision letter (RSOS-201927.R0)

Dear Mr Razik

On behalf of the Editors, we are pleased to inform you that your Manuscript RSOS-201927 "Non-kin adoption in the common vampire bat" has been accepted for publication in Royal Society Open Science subject to minor revision in accordance with the referees' reports. Please find the referees' comments along with any feedback from the Editors below my signature.

Please submit your revised manuscript and required files (see below) no later than 7 days from today's (ie 06-Jan-2021) date. Note: the ScholarOne system will 'lock' if submission of the revision is attempted 7 or more days after the deadline. If you do not think you will be able to meet this deadline please contact the editorial office immediately.

on behalf of Dr Dieter Lukas (Associate Editor) and Kevin Padian (Subject Editor)
openscience@royalsociety.org

Associate Editor Comments to Author (Dr Dieter Lukas):

Associate Editor: 1

Comments to the Author:

I agree with the reviewers that this is an interesting contribution that should be part of the published literature for others to build upon. However, before publication, the reviewers highlight important points where the authors should (i) provide additional details to follow the arguments and (ii) more clearly express the limitations of what can be inferred from this study given that it is based on a rare behavior in a captive colony. In the discussion of the limitations, it might be worth considering why adoption behavior might generally be rare or rarely observed. For example, is maternal mortality higher among vampire bats than other long-lived bat species? All these points are however minor, and I hope the authors will be able to easily address these in their revisions.

Reviewer comments to Author:

Reviewer: 1

Comments to the Author(s)

In this short communication paper Razik investigated the non-kin adoption of an orphaned vampire bat (*Desmodus rotundus*) by an adult female (BD) after his mother (Lilith) death 19 days after giving birth in a captive bat colony in the SRI, Panama. Using three surveillance cameras, he collected 652 observation hours over four months to describe patterns of allogrooming and food sharing between the 23 adult females before and after the adoption event. After Lilith's died, the orphaned pup received allomaternal care from BD, and this infant survived until the end of the experiment. He concludes that the observation lend support to the hypothesis non-kin adoption can be motivated by a history of cooperative interactions.

This is a very well written paper on the proximal and evolutive causes of the of the allomaternal care and adoption of orphaned tropical bats. In the Introduction, Methods, and Results sections the author provided the necessary information to understand and study phenomenon (i.e. the role of inter-bat cooperative interactions on the allomaternal).

Certainly this work will be an interesting contribution on the topic. However, my main concern is that the extrapolation power of this study to the entire species is poor, because the lack of replicates (i.e. the number of allomaternal adoption events provided by the author is 1). Even, it is not clear for me if the main findings of this study could be extrapolated to free-ranging bats. Furthermore, most literature provide by the author in the discussion is based in captive bat colonies, then, allomaternal care may be a particular behavior of captive bats but not a 'natural behavior' of *Desmodus rotundus*. I understand that this type of events is, probably, very rare even in captivity, but the author must recognize the study limitations in the text and then, down the tone of the discussion.

Other issues

p.4, lines 7-24: Even when the author mention some appropriate statistical procedures to test the hypothesis, no statistical vales are provided in the Results section. This is a little strange and do not benefit the Result section, which result a largely descriptive section. Then, I author must indicate the statistical values of the tests in the text.

p.4, lines 32, 34, 35, 50, 52: The Figure 1 is not available in the proof provided by the editor. Only the Figure legend is provided even when this figure was important to understand the main findings.

p.5, lines 5-12: In this first discussion paragraph the authors repeat the main results. For me this paragraph must be removed. The discussion section must be focused in providing reasonable explanations to the main results, but not in synthetize the main results.

line 24: replace 'adoptive female' by 'adoptive mother'. The term 'adoptive female' has no biological sense. The same problem also is present in the Figure 1 according the R scripts provided.

p.5, lines 46-60: in this paragraph the author provided some reasons explaining why vampire bats are 'interesting comparative case' for studying the biological mechanism behind adoption. However, I suggest that this paragraph (or part of it) must be relocated to the introduction because this information do not contribute to explain the main results of this study.

p.6, lines 36-37: Because the reasons I have mentioned above, authors must down up the tone of this affirmation.

Reviewer: 2

Comments to the Author(s)

This paper provides a detailed description of an interesting evolutionary problem – why female vampire bats adopt the offspring of unrelated females. A weakness of the study is that it details just one case of this unusual behaviour (which has been reported before) in a captive environment, but the behaviour is very well documented, the paper is clearly written and uses appropriate statistical analyses. I therefore believe it still makes a valuable contribution to the literature, and with a bonus that it was also enjoyable to read!

Major comments:

1. Could this adoptive behaviour just be an abnormal behaviour induced by captivity and not observed at all in the wild? This is discussed to some extent L30-38 on page 5 but I think it is worth emphasizing this as a potential caveat in later parts of the discussion covering how this adoptive behaviour may have evolved (particularly L16-29 of page 6).

2. I was surprised by the decision to permute both between time periods (by randomizing the direction of change) and between adoptive and non-adoptive females. My intuition is that this would mean a significant difference between the observed effect and the null effect could be driven either by adopter/non-adopter differences or by changes over time (e.g. all females provide more care in the period after maternal loss or just that food sharing and grooming increase over time as relationships strengthen between individuals that have spent more time in captivity together). By permuting both things we cannot tell which is driving this difference. It could be worth running this as two separate permutations, one asking does food sharing and grooming significantly increase between periods and another asking do they increase more between periods for the supposed adopter compared to other females?

Minor comments:

L27 p.2 and L15 p.13 – I was initially confused by what previously unfamiliar meant until I reached the methods. Is there a better way of phrasing this to making it clearer what is meant early on? E.g. previously unfamiliar females placed together within a captive colony.

L58 p.2 - Is it worth discussing this from the perspective of the individual being adopted as well as the individual doing the adopting? Could there be selection for traits in young bats that encourage alloparental care? Im thinking about this from a parent-offspring conflict sort of perspective.

L55 p.4 - 'BD adopted lilith's pup' – it seems a bit too soon to say this. Might be better to first introduce the data and analyses that support this and then state it.

L58 p.4 - I think it needs to be made clearer that these 'expected increases' are 95% confidence intervals based on the null models. I ended up going through the code to double check.

Reviewer: 3

Comments to the Author(s)

This interesting manuscript explores an exciting observation of adoption (including lactation in a previously non-lactating adult female) of a putatively unrelated pup. Adoption in non-human animals is poorly described in the literature especially amongst such well-studied mammals such as the vampire bats described here. To this end, I think this paper will be of broad interest. It is very well written and organized and the length seems appropriate. The biggest concern I have is how we can be certain that the 2 adult females are not related. Any evidence that could be provided to clarify this- even just greater explanations regarding the degree of relatedness normally observed at the same roost would be helpful. That said, some concerns may remain regarding how females disperse through the landscape and if low levels of relatedness at a single roost could be explained by females leaving at adulthood. If that were the case bats captured 100 km apart may be more likely to be related. Clarification in the text would at least help alleviate such concerns. The observations around lactation in a previously non-reproductive individual could be highlighted a bit more. Limitations of a single observed adoption (in this study) could be addressed in the discussion. Overall, I found this paper a fun and interesting read and I'm sure that others will feel similarly.

General comments.

Please explain how you can be certain that the two adult females are unrelated.

Please clarify (unless I am mistaken) that you do not know exactly when nor why BD was able to start lactating (re it was not truly 'spontaneous'). The older literature sometimes uses terms that have not withstood the test of time so I would see if there is a new and more appropriate way to describe this.

Would this observation of a single adoption have adaptive significance considering the Carter and Wilkinson labs raw number of hours watching vampire bats? It seems it must be extremely rare. Thus, it may be a fluke. That said it is fascinating and certainly thought-provoking.

Nevertheless, you may wish to talk about this caveat as well.

Specific comments by lines unless otherwise noted.

Abstract.

22. Specify how it 'enhances parental care.' Do you mean they learn to be better parents?

23. Rephrase for clarity: "...adoption is the strength of previous cooperative..."

26. Specify that the sex of the pup is female.

29. Explain what 'feeding' means here. Re. Blood sharing.

Introduction.

Page 2 of 8.

You only list indirect and nonadaptive benefits. Please also list 'adaptive' / 'non-indirect fitness' given the focus of your study.

44-51. I was surprised that humans were not discussed a little bit here. In an effort to indicate that evolutionary theory applies to humans and non-humans alike I would list them as one of your case examples here. I suspect some papers are available re- adoption amongst the Ache or Hazda.

54 - 55. Explain 'extreme attraction.' Maybe mention here that hormone levels may lead to displace parental caregiving.

Page 3 of 8.

5 - 9 Does this include humans (citations 9 - 11)?

19 - 21. Is this hypothesis that adoption is based in part on cooperative interaction something that has been discussed in bats before? If not, I would highlight here that this is the first time this idea has been postulated.

13, 14. Unrelated and previously unfamiliar. This idea is introduced here without any explanation. Fortunately, below you explain the distance between the sites (34,35). However, the reason they are believed to be unrelated should be explained earlier as it is key for the premise of your paper. That said, distance does not seem so large as to preclude them from being related. Towards the end of the manuscript, you mentioned the low r -value in vampire bats at the same roost. That information should also be given earlier (here is) to clarify that within a roost, vampire bats are not closely related. All that said, this is a key point that would ideally be addressed if you had any genetic material to verify the degree of relatedness between Lilith and BD. This also presents my only scientific concern with the manuscript and an important caveat to the study.

Methods general comments.

Where are Lilith and BD from (re - the 2 sites?)

53 - 56. Fasting is a (somewhat) unique state used experimentally in your study for obvious reasons. Do you think that this could have elicited the adoption behavior by inducing greater amounts of food sharing and thus adoption was a byproduct of greater interactions overall (maybe that increases 'priming' for adoption)? Could this explain why Wilkinson didn't observe adoption in naturally occurring free-living bats he studied?

Page 4 of 8.

3 - 6. I would reference your figure here.

Results.

References to figure one in the first paragraph made me think there was an illustration depicting how BD was not noticeably lactating. If you do not have any such pictures (which would be a nice addition if you did) I would move the reference to figure 1, so as to avoid this confusion.

31. Also in reference to the figure, I suggest adding important events such as Lilith meeting BD, Lilith being removed from the colony, and Lilith being returned to the colony. See my note in my reference to the figure below.

The first paragraph of results.

I was wondering if BD is not just a 'giver' perhaps with abnormally high oxytocin levels? Below you refer to the fact that she did not provide noticeably high amounts of care for other bats. However, it is unclear when that observation was true relative to her adopting the pup. I think this needs clarification and reference to a key time point on your figure 1.

These second, third, and fourth paragraphs seem out of order. For example, you talk about Lilith giving birth in the third paragraph after you've talked about her decline after giving birth earlier on.

55 - 56. "On this day BD was found to be lactating." I would clarify in the text that she did not likely spontaneously start lactating exactly on the 28th of August. Unfortunately, you cannot rule out that she gradually started beforehand which, makes more sense physiologically. If you can say for certain, elaborate a bit on how you know or how you 'first noticed her lactating on...' Discuss this caveat and how this relates to when we may have started supplementing the pup with milk. Please note on the next page line 24 through 25 you also mention spontaneously lactating. I would call this something else. For example, you could call it induced lactation or similar. Again, spontaneous would imply no hormonal nor behavioral priming which you have no evidence of (hormonal) and possible behavioral priming given earlier bouts of grooming.

Fourth paragraph line 60 to the end of page.

Here you refer to BD not being the top groomer nor food donor. However, it is unclear to me when this statement is for the whole study, before or after she adopts the pup. Please clarify.

Page 5 of 8 continued.

46 - 49. How often does a single female have more than one pup in a single year? If it is not particularly common, she may benefit more from taking care of a possible help her relative to trying to squeeze in yet another pup in a year. I'm curious which would be the more energetically beneficial strategy.

50 - 51. Here is where you discuss the low kinship value. Again, this ought to be mentioned much earlier as it is unclear how you know that these 2 females are unrelated.

Also, this paragraph.

I think you could highlight just how enormously expensive 9 months of lactation would truly be to help illustrate the cost of this unique behavior.

Page 6 of 8.

2 - 3. Could this occur vice versa where maternal care was co-opted to lead to the origins of food sharing? I suspected this could well be what happened in early humans.

Line 8.

Your discussion really focuses on the social interactions of the 2 adult females but here you discuss interactions with the orphan prior to the mother's death. Can you carefully parse these ideas up and either only focus on one or make sure you discuss the importance of interactions at both levels?

Paragraph lines 15 through 29.

I did not quite follow what was meant by "parental guided matchmaking, sexual imprinting" and to some extent even "deceit." If an adoptive mother is actually feeding a pup then she is truly giving the pup some benefit- so are you suggesting that the pup is expected to help later on but ultimately does not? Clarify.

Figure 1A timeline.

I would add important key events onto this figure. For example: When did the 2 adults meet? When was Lilith removed? When was Lilith returned? You could also add on hatched bars for the duration of lactation to highlight the period over which this is occurring for each of the 2 mothers.

For consistency sake I would use the names in parentheses on this figure as well.

Figures 1 B etc.

I would give a number in parenthesis dashed grey lines to highlight the focus possible number of interactions. This gets back to my previous question regarding whether BD isn't just a 'giver' and if her behavior changed after she adopted the pup.

Reviewer: 4

Comments to the Author(s)

I have reviewed Rezik et al's manuscript titled 'Non-kin adoption in the common vampire bat'. I thought this was a well-written, succinct manuscript. The article describes a single incidence of offspring adoption in the common vampire bat. This behavior is of interest because it is exceptionally rare, especially among non-kin. Suspected adoption in common vampires was reported in a prior manuscript, but this is the first to make detailed observations that confirm the behavior.

I think the introduction was well written, touching on all appropriate background, and the methods were clear. The use of polynomial regression fitting was a reasonable approach to characterizing the data.

In the results, the timeline was a bit awkward. In the paragraph starting line 41, activities after Lilith's death were described. Then, the paragraph that starts on line 48 jumps back to describing when Lilith gave birth. To me, this would have better flow if the sentence starting 'On 9 August 2019..' was removed and the next started 'In the weeks following parturition,...'. The date of birth was described previously in the methods.

I thought the figures nicely portrayed the patterns described.

In the conclusion, the authors state that social interactions strongly impact non-kin adoption. While I agree that is likely, I strongly recommend that authors tone this down, as it is impossible to draw strong conclusions from a single adoption event.

Overall, I think this paper will be of interest to bat biology and the organismal biology community. The biggest problem with this manuscript is that it is based on a single observation of adoption. Thus, the authors and readers should not be confident that the observed behaviors would be typical of adoption in this species, and it remains unclear how common this behavior is. In the past, journals such as the Journal of Mammalogy had a 'Notes' section for important natural history observations, such as this one. Few journals will publish this type of natural history observation based on a single observation – it's up to the editors if natural history observations have a home in the Royal Society Open Science. I think natural history observations are important, and it would be supportive of Open Science publishing this manuscript if it meets their standards.

===PREPARING YOUR MANUSCRIPT===

If you have been asked to revise the written English in your submission as a condition of publication, you must do so, and you are expected to provide evidence that you have received language editing support. The journal would prefer that you use a professional language editing service and provide a certificate of editing, but a signed letter from a colleague who is a native speaker of English is acceptable. Note the journal has arranged a number of discounts for authors

using professional language editing services
(<https://royalsociety.org/journals/authors/benefits/language-editing/>).

===PREPARING YOUR REVISION IN SCHOLARONE===

-- If you have uploaded ESM files, please ensure you follow the guidance at <https://royalsociety.org/journals/authors/author-guidelines/#supplementary-material> to include a suitable title and informative caption. An example of appropriate titling and captioning may be found at https://figshare.com/articles/Table_S2_from_Is_there_a_trade-

off_between_peak_performance_and_performance_breadth_across_temperatures_for_aerobic_sc
ope_in_teleost_fishes_/3843624.

Author's Response to Decision Letter for (RSOS-201927.R0)

See Appendix A.

Decision letter (RSOS-201927.R1)

Dear Mr Razik,

It is a pleasure to accept your manuscript entitled "Non-kin adoption in the common vampire bat" in its current form for publication in Royal Society Open Science.

on behalf of Dr Dieter Lukas (Associate Editor) and Kevin Padian (Subject Editor)
openscience@royalsociety.org

Appendix A

Associate Editor Comments to Author (Dr Dieter Lukas):

Associate Editor: 1

Comments to the Author:

I agree with the reviewers that this is an interesting contribution that should be part of the published literature for others to build upon. However, before publication, the reviewers highlight important points where the authors should (i) provide additional details to follow the arguments and (ii) more clearly express the limitations of what can be inferred from this study given that it is based on a rare behavior in a captive colony. In the discussion of the limitations, it might be worth considering why adoption behavior might generally be rare or rarely observed. For example, is maternal mortality higher among vampire bats than other long-lived bat species? All these points are however minor, and I hope the authors will be able to easily address these in their revisions.

Thank you for the constructive comments! We are glad that the paper was well received, and we thank the reviewers and editor for their meticulous, insightful comments which have greatly improved the clarity of the manuscript. Below you will find our responses to reviewer comments in bold, with original and revised text in italics.

Best,

Imran Razik, Bridget Brown, Rachel Page, and Gerald Carter

Reviewer comments to Author:

Reviewer: 1

Comments to the Author(s)

In this short communication paper Razik investigated the non-kin adoption of an orphaned vampire bat (*Desmodus rotundus*) by an adult female (BD) after his mother (Lilith) death 19 days after giving birth in a captive bat colony in the SRI, Panama. Using three surveillance cameras, he collected 652 observation hours over four months to describe patterns of allogrooming and food sharing between the 23 adult females before and after the adoption event. After Lilith's died, the orphaned pup received allomaternal care from BD, and this infant survived until the end of the experiment. He concludes that the observation lend support to the hypothesis non-kin adoption can be motivated by a history of cooperative interactions.

This is a very well written paper on the proximal and evolutive causes of the of the allomaternal care and adoption of orphaned tropical bats. In the Introduction, Methods, and Results sections the author provided the necessary information to understand and study phenomenon (i.e. the role of inter-bat cooperative interactions on the allomaternal).

Certainly this work will be an interesting contribution on the topic. However, my main concern is that the extrapolation power of this study to the entire species is poor, because the lack of replicates (i.e. the number of allomaternal adoption events provided by the author is 1). Even, it is not clear for me if the main findings of this study could be extrapolated to free-ranging bats. Furthermore, most literature provide by the author in the discussion is based in captive bat colonies, then, allomaternal care may be a particular behavior of captive bats but not a 'natural behavior' of *Desmodus rotundus*. I understand that this type of events is, probably, very rare even in captivity, but the author must recognize the study limitations in the text and then, down the tone of the discussion.

We thank the reviewer for these kind words and agree that a critical limitation to the study is the small sample size and the fact that this observation occurred in captivity, therefore making extrapolation to wild behavior difficult. We now acknowledge and emphasize these points more clearly in the revised discussion and conclusion.

Revised text in discussion:

It is important to note that the few observed cases of adoption in vampire bats all occurred in captivity. During more than 400 hours of field observation, Wilkinson [25,30,32,34] observed allogrooming of vampire bat pups, but he never observed allonursing. Adoption would be difficult to observe in wild colonies because orphaned pups are uncommon. In our captive colony, it is also interesting to note that another female, BSCS, noticeably increased its allogrooming rate to the pup after the mother's death (see Figure 2). Both BD and BSCS had previously lived in captivity for almost two years during another study [27,28]. We therefore speculate that both captivity and death of the mother might increase the probability of adoption in vampire bats. However, the overall probability of adoption is difficult to estimate because observations of orphaned pups are rare.

Revised text in conclusion:

Observations of non-kin pup adoption in captive common vampire bats are consistent with the hypothesis that non-kin adoption is a byproduct of proximate cognitive and neuroendocrine mechanisms that are crucial for parental care and triggered by unusual circumstances. Our observations also suggest that adoption is influenced by the history of cooperative interactions between mother, offspring, and adopter, although more evidence is needed to test this hypothesis. The probability of non-kin adoption in vampire bats and whether it affects the fitness of adopters remains unclear, especially outside of captive conditions, but we can make two testable predictions. If non-kin adoption in vampire bats is adaptive, we predict that female pups should be adopted more often than male pups because females are philopatric. If female vampire bats are more motivated to help the offspring of more closely bonded partners, then vampire bat pups should be biased towards inheriting the cooperative relationships of their mothers, i.e. social inheritance of network ties [49].

Other issues

p.4, lines 7-24: Even when the author mention some appropriate statistical procedures to test the hypothesis, no statistical values are provided in the Results section. This is a little strange and do not benefit the Result section, which result a largely descriptive section. Then, I author must indicate the statistical values of the tests in the text.

We have revised the results which now provides statistical values:

After this day, BD increased its rate of allogrooming the pup by 47 seconds/hour (s/h) and food sharing with the pup by 15 s/h (Figure 2), which was greater than the 20 other bats (nonparametric rank test $p = 0.048$ for both allogrooming and food sharing; parametric outlier test, allogrooming: $G = 3.0$, $p = 0.007$, food sharing: $G = 4.0$, $p < 0.0001$).

p.4, lines 32, 34, 35, 50, 52: The Figure 1 is not available in the proof provided by the editor. Only the Figure legend is provided even when this figure was important to understand the main findings.

We apologize that the figure was unavailable. The figure was submitted and should have been included at the end of the PDF document. We include it again with this revision.

p.5, lines 5-12: In this first discussion paragraph the authors repeat the main results. For me this paragraph must be removed. The discussion section must be focused in providing reasonable explanations to the main results, but not in synthesize the main results.

We no longer restate or synthesize the main results, and instead only briefly summarize the take home message. We do feel it is necessary to summarize what we believe the results mean.

Revised text:

We observed the complete social history leading up to a case of non-kin adoption between unrelated common vampire bats. In captivity, a mother gave birth then died after several weeks. During this time, another previously unfamiliar and unrelated female responded to changes in need of both the mother and the pup (Figure 1). After the mother's death, the female increased the rate at which it allogroomed and regurgitated food to the orphaned pup. We also noticed that this female was providing the pup with milk. This increase in helping behavior was uniquely directed to the pup (in comparison to other juveniles or adults in the colony), and no other female helped the orphaned pup to nearly the same extent (Figure 2).

line 24: replace 'adoptive female' by 'adoptive mother'. The term 'adoptive female' has no biological sense. The same problem also is present in the Figure 1 according the R scripts provided.

We agree "adoptive female" is confusing. For clarity, we have changed "adoptive female" to "adopter" or "adopting female" throughout the text and figures, because we would prefer to only use the term "mother" in the biological sense.

p.5, lines 46-60: in this paragraph the author provided some reasons explaining why vampire bats are 'interesting comparative case' for studying the biological mechanism behind adoption. However, I suggest that this paragraph (or part of it) must be relocated to the introduction because this information do not contribute to explain the main results of this study.

We have revised this text. In the original text, we incorrectly implied that this paragraph is about the reasons to do our study, which is why it seems to belong in the introduction, but it is actually about reasons to do future work, which is why it belongs in the discussion. We have revised the text to clarify this. Also, our claim that vampire bats make an interesting comparative case for studying non-kin adoption is based directly off the results, which can only be fully synthesized in the discussion.

p.6, lines 36-37: Because the reasons I have mentioned above, authors must down up the tone of this affirmation.

We agree with the reviewer and have revised the text with more conservative language. Revised text:

Observations of non-kin pup adoption in captive common vampire bats are consistent with the hypothesis that non-kin adoption is a byproduct of proximate cognitive and neuroendocrine mechanisms that are crucial for parental care and triggered by unusual circumstances. Our observations also suggest that adoption is influenced by the history of cooperative interactions

between mother, offspring, and adopter, although more evidence is needed to test this hypothesis. The probability of non-kin adoption in vampire bats and whether it affects the fitness of adopters remains unclear, especially outside of captive conditions, but we can make two testable predictions. If non-kin adoption in vampire bats is adaptive, we predict that female pups should be adopted more often than male pups because females are philopatric. If female vampire bats are more motivated to help the offspring of more closely bonded partners, then vampire bat pups should be biased towards inheriting the cooperative relationships of their mothers, i.e. social inheritance of network ties [49].

Reviewer: 2

Comments to the Author(s)

This paper provides a detailed description of an interesting evolutionary problem – why female vampire bats adopt the offspring of unrelated females. A weakness of the study is that it details just one case of this unusual behaviour (which has been reported before) in a captive environment, but the behaviour is very well documented, the paper is clearly written and uses appropriate statistical analyses. I therefore believe it still makes a valuable contribution to the literature, and with a bonus that it was also enjoyable to read!

Thank you for the feedback! We are glad to hear that the paper was an enjoyable narrative. We share the disappointment that we have just a single observation of the adoption behavior, but we are excited to have one of the most complete descriptions of the shared social history between the individuals engaged in this rare event. In our revised manuscript, we have more explicitly addressed the limitations of our study.

Major comments:

1. Could this adoptive behaviour just be an abnormal behaviour induced by captivity and not observed at all in the wild? This is discussed to some extent L30-38 on page 5 but I think it is worth emphasizing this as a potential caveat in later parts of the discussion covering how this adoptive behaviour may have evolved (particularly L16-29 of page 6).

Yes, and we agree. We added text to cover this point in the final paragraph of the discussion. The final paragraph of the discussion now states:

Adoption can be explained completely, or in part, as a byproduct of normal parental care. For example, if parental behaviors are triggered by a common set of neuroendocrine mechanisms or stimuli (e.g. [44]), then the same traits that lead to adaptive behavior under typical circumstances could also cause non-adaptive adoptions under the rare circumstance of an orphaned infant in dire need. Moreover, the generally atypical circumstances associated with captive conditions may sufficiently increase the likelihood of non-kin adoption, which may not occur often or at all in the wild. Some authors have suggested ways in which non-kin adoption might be adaptive for the adopter, including reciprocity [13,45], 'match-making' between biological and adopted young to form compatible mating pairs [46], or kinship deceit, by which adopters exploit kin-recognition heuristics and deceive adopted young into a false perception of kinship, thereby causing these young to later help at the nest [47]. Most, if not all, adaptive explanations predict that non-kin adoption of orphaned infants should be biased towards the philopatric sex, but this pattern has not been clearly shown [11–13]. In the case of vampire bats, it is also hard to explain why a female would invest in an asymmetric helping relationship with an infant over a potentially reciprocal relationship with an adult female [48],

but these arguments do not eliminate the possibility that adopters may eventually receive social benefits from adopted individuals.

2. I was surprised by the decision to permute both between time periods (by randomizing the direction of change) and between adoptive and non-adoptive females. My intuition is that this would mean a significant difference between the observed effect and the null effect could be driven either by adopter/non-adopter differences or by changes over time (e.g. all females provide more care in the period after maternal loss or just that food sharing and grooming increase over time as relationships strengthen between individuals that have spent more time in captivity together). By permuting both things we cannot tell which is driving this difference. It could be worth running this as two separate permutations, one asking does food sharing and grooming significantly increase between periods and another asking do they increase more between periods for the supposed adopter compared to other females?

After much thought, we realized that the permutation test is difficult to interpret. We instead present the data using the following Figure. As you can see, the raw data speak for themselves.

Figure 2. Increases in rates of allogrooming and food sharing from 21 adult females to the orphaned pup before and after the mother's death. The "adopter" increased its allogrooming and food-sharing rates to the pup more than any other female. The allogrooming and food-sharing rates are shown for 21 bats, but most of the slopes are not visible because many of the rates are zero both before and after the mother's death.

Now the reader can clearly see the difference between the behavior of the adopter versus other females. To test the null hypotheses, we replaced the permutation tests with a simple exact p-value based on ranks and an outlier test on the log-transformed rates. Conclusions remain the same but are easier to understand.

Revised methods:

To see if the increase in BD's helping behavior was greater than expected by chance, we measured the mean change in allogrooming and food-sharing rates towards the pup before versus after the

mother's death for all 21 adult females that were present in the colony during both periods (one female died before the mother gave birth). We then calculated the exact probabilities that BD would have the highest increases by chance (nonparametric test) and we used Grubb's Test to detect if BD's increase in allogrooming and food-sharing towards the pup was an outlier (parametric test). For Grubb's test, we used $\log(\text{seconds}+1)$ to increase normality of the allogrooming and food-sharing rates before calculating the mean change. We conducted all analyses in R version 3.6.1 [31].

Revised results:

After this day, BD increased its rate of allogrooming the pup by 47 seconds/hour (s/h) and food sharing with the pup by 15 s/h (Figure 2), which was greater than the 20 other bats (nonparametric rank test $p = 0.048$ for both allogrooming and food sharing; parametric outlier test, allogrooming: $G = 3.0$, $p = 0.007$, food sharing: $G = 4.0$, $p < 0.0001$). BD became the pup's highest ranked groomer and food donor, but BD was not the top groomer or food donor for any of the seven other juvenile bats in the colony. BD groomed the pup an average of 52 s/h but groomed other juveniles only 0 to 4 s/h. BD fed the pup an average of 13 s/h but fed other juveniles 0 to 1 s/h. BD was still nursing the pup when we finished observations on 14 October 2019.

Minor comments:

L27 p.2 and L15 p.13 – I was initially confused by what previously unfamiliar meant until I reached the methods. Is there a better way of phrasing this to making it clearer what is meant early on? E.g. previously unfamiliar females placed together within a captive colony.

We agree that more clarity early on would be helpful. The revised abstract now states:

*...two adult female common vampire bats (*Desmodus rotundus*) that were captured from distant sites (340 km apart) and introduced to one another in captivity.*

Revised text in the introduction now reads:

*We tracked the allogrooming and food-sharing relationships among 23 adult female common vampire bats (*Desmodus rotundus*) before and after one female adopted the orphaned pup of another female bat that was unrelated and previously unfamiliar. The two adult females, named 'BD' and 'Lilith', first met in captivity after being captured from sites 340 km apart.*

L58 p.2 - Is it worth discussing this from the perspective of the individual being adopted as well as the individual doing the adopting? Could there be selection for traits in young bats that encourage alloparental care? I'm thinking about this from a parent-offspring conflict sort of perspective.

Yes, there are certainly traits in young bats that would encourage alloparental care such as isolation calls and attempting to nurse from other females; however, most or all of these traits would also encourage maternal care, so we think speculation on this topic is beyond the scope of this paper. However in the discussion we discuss the pup's perspective when we mention that "The survival of offspring would appear to benefit from being adopted [1], but this is not always evident, especially among populations with high mortality [1,12,13]."

L55 p.4 - 'BD adopted lilith's pup' – it seems a bit too soon to say this. Might be better to first introduce the data and analyses that support this and then state it.

We agree that this was mentioned too early and is probably not needed. We removed this text from the paragraph.

L58 p.4 – I think it needs to be made clearer that these ‘expected increases’ are 95% confidence intervals based on the null models. I ended up going through the code to double check.

We have deleted this text, because we now perform a simpler statistical test and show the data directly.

Reviewer: 3

Comments to the Author(s)

This interesting manuscript explores an exciting observation of adoption (including lactation in a previously non-lactating adult female) of a putatively unrelated pup. Adoption in non-human animals is poorly described in the literature especially amongst such well-studied mammals such as the vampire bats described here. To this end, I think this paper will be of broad interest. It is very well written and organized and the length seems appropriate. The biggest concern I have is how we can be certain that the 2 adult females are not related. Any evidence that could be provided to clarify this- even just greater explanations regarding the degree of relatedness normally observed at the same roost would be helpful. That said, some concerns may remain regarding how females disperse through the landscape and if low levels of relatedness at a single roost could be explained by females leaving at adulthood. If that were the case bats captured 100 km apart may be more likely to be related. Clarification in the text would at least help alleviate such concerns. The observations around lactation in a previously non-reproductive individual could be highlighted a bit more. Limitations of a single observed adoption (in this study) could be addressed in the discussion. Overall, I found this paper a fun and interesting read and I’m sure that others will feel similarly.

We are really glad you enjoyed the manuscript. Thank you for the feedback!

General comments.

Please explain how you can be certain that the two adult females are unrelated.

We highlighted the important detail that the bats were captured at sites 340 km apart.

The probability that two female vampire bats would be kin if they are randomly sampled from distant sites hundreds of kilometers apart is effectively zero (analogous to the probability that two randomly sampled humans from distant cities are in the same family). Vampire bats are among the most common species in Panama, females are philopatric, and they have a mean home range size of 35 hectares (Wilkinson 1985). The vast distance between the capture sites is actually better evidence of the females being unrelated than we would get from typical genetic data. Even if genetic data said they were related, it would still be far more likely that the genetic estimates were flawed than the two bats are closely related.

Please clarify (unless I am mistaken) that you do not know exactly when nor why BD was able to start lactating (re it was not truly ‘spontaneous’). The older literature sometimes uses terms that have not withstood the test of time so I would see if there is a new and more appropriate way to describe this.

We agree with the reviewer's recommendation for cautious wording here. This is correct, we are not sure exactly when BD began to lactate. Along with other reviewer suggestions, we have now restructured the results section and added more concrete language when speaking of how and when we were able to determine that BD was lactating.

The methods section now states:

Allomaternal care from BD included grooming, regurgitated food sharing, and nursing, which we confirmed by manually expressing milk from the nipple on the day Lilith died.

The results text now states:

At this time, BD was not noticeably lactating.

During this time, observations from video footage suggested that BD began to nurse the pup, which seemed to increase gradually.

On this day, we observed that BD was lactating and providing milk to the pup.

Would this observation of a single adoption have adaptive significance considering the Carter and Wilkinson labs raw number of hours watching vampire bats? It seems it must be extremely rare. Thus, it may be a fluke. That said it is fascinating and certainly thought-provoking. Nevertheless, you may wish to talk about this caveat as well.

We completely agree that the adaptive significance (if any) and the rarity of this behavior is unknown. We have revised the text to make this clearer (see revisions above and below). It is important to note that adoptions can only occur if a pup is orphaned and observations of orphaned pups are rare. If a pup was adopted, it would also be difficult to know that unless the observer witnessed the mother's death. We cannot speak directly for Wilkinson, but Carter has spent hundreds of hours watching vampire bats and did not see a case where a pup was orphaned and could have been adopted. We would suspect something similar for Wilkinson.

Specific comments by lines unless otherwise noted.

Abstract.

22. Specify how it 'enhances parental care.' Do you mean they learn to be better parents?

If a female is more responsive to offspring need, that would lead to more frequent care behaviors.

Revised text now reads:

Although adoption may be best explained as a byproduct of adaptive traits that enhance parental care or promote the development of parental skills...

23. Rephrase for clarity: "...adoption is the strength of previous cooperative..."

Revised:

...one factor that is possibly important for the likelihood of adoption is the history of cooperative interactions between the mother, adopted offspring, and adopter.

26. Specify that the sex of the pup is female.

Fixed

29. Explain what 'feeding' means here. Re. Blood sharing.

Fixed

Introduction.

Page 2 of 8.

You only list indirect and nonadaptive benefits. Please also list 'adaptive' / 'non-indirect fitness' given the focus of your study.

Revised text:

In some cases, alloparental care towards either kin or non-kin could also result in direct fitness benefits for the helper (e.g. group augmentation [5,6]).

44-51. I was surprised that humans were not discussed a little bit here. In an effort to indicate that evolutionary theory applies to humans and non-humans alike I would list them as one of your case examples here. I suspect some papers are available re- adoption amongst the Ache or Hazda.

Indeed, we agree that it would be useful to mention humans here. We added the following text:

In humans, adoption of kin or non-kin children is well-described in both traditional and modern industrial societies around the world [19].

54 - 55. Explain 'extreme attraction.' Maybe mention here that hormone levels may lead to displace parental caregiving.

Thank you for pointing this out. We have changed this language:

In several primate species, females often appear interested in handling others' infants and may even try to kidnap infants from other group members, which may be associated with reproductive seasonality or adolescence [13,20,21].

Page 3 of 8.

5 - 9 Does this include humans (citations 9 – 11)?

We now say "non-human primates".

19 - 21. Is this hypothesis that adoption is based in part on cooperative interaction something that has been discussed in bats before? If not, I would highlight here that this is the first time this idea has been postulated.

It has not been discussed in bats before, but this is not surprising, given how little is known about cooperative interactions in other bat species. We are not sure if this is the first time the idea has been postulated.

13, 14. Unrelated and previously unfamiliar. This idea is introduced here without any explanation. Fortunately, below you explain the distance between the sites (34,35). However, the reason they are believed to be unrelated should be explained earlier as it is key for the premise of your paper. That said, distance does not seem so large as to preclude them from being related. Towards the end of the manuscript, you mentioned the low r-value in vampire bats at the same roost. That information should also be given earlier (here is) to clarify that within a roost, vampire bats are not closely related. All that said, this is a key point that would ideally be addressed if you had any genetic material to verify the degree of relatedness between Lilith and BD. This also presents my only scientific concern with the manuscript and an important caveat to the study.

This issue has been brought up and addressed by other reviewers above. We clarified that the bats are from sites 340 km on opposite ends of Panama, so they cannot be familiar or closely related.

Methods general comments.

Where are Lilith and BD from (re - the 2 sites?)

See above.

53 - 56. Fasting is a (somewhat) unique state used experimentally in your study for obvious reasons. Do you think that this could have elicited the adoption behavior by inducing greater amounts of food sharing and thus adoption was a byproduct of greater interactions overall (maybe that increases 'priming' for adoption)? Could this explain why Wilkinson didn't observe adoption in naturally occurring free-living bats he studied?

Possibly! To some extent this is exactly why we consider the hypothesis that need-based cooperative interactions affect the likelihood of non-kin adoption. The few analyses we used in this study examine all cooperative interactions between Lilith, Lilith's pup, and BD. When Lilith was fasted, Lilith generally received food donations from BD. Had we not fasted Lilith, it is possible that we would not have seen the same patterning and quality of interactions between BD, Lilith, and Lilith's pup. That is, the adoption may have been less likely. Nevertheless, this is only speculation. Another interesting question is why Lilith rarely appeared to share food with BD, given that their grooming relationship was highly symmetrical.

Page 4 of 8.

3 - 6. I would reference your figure here.

Fixed.

Results.

References to figure one in the first paragraph made me think there was an illustration depicting how BD was not noticeably lactating. If you do not have any such pictures (which would be a nice addition if you did) I would move the reference to figure 1, so as to avoid this confusion.

Unfortunately, we did not take a picture at the time of capture or when we checked if the bat was lactating. To avoid confusion, the original sentence (page 4, lines 31 – 32) no longer includes the parenthetical reference to Figure 1A.

31. Also in reference to the figure, I suggest adding important events such as Lilith meeting BD, Lilith being removed from the colony, and Lilith being returned to the colony. See my note in my reference to the figure below.

We have made some small edits to the timeline in Figure 1A to improve clarity:

The first paragraph of results.

I was wondering if BD is not just a ‘giver’ perhaps with abnormally high oxytocin levels? Below you refer to the fact that she did not provide noticeably high amounts of care for other bats. However, it is unclear when that observation was true relative to her adopting the pup. I think this needs clarification and reference to a key time point on your figure 1.

We now address this in our results. Revised text:

BD became the pup’s highest ranked groomer and food donor, but BD was not the top groomer or food donor for any of the seven other juvenile bats in the colony. BD groomed the pup an average of 52 s/h but groomed other juveniles only 0 to 4 s/h. BD fed the pup an average of 13 s/h but fed other juveniles 0 to 1 s/h. BD was still nursing the pup when we finished observations on 14 October 2019.

Among all bats, BD was not exceptionally cooperative. Overall, BD ranked 5th for having the highest allogrooming and food-sharing rates towards other adult females. Besides Lilith and her pup, BD was also the top groomer for only one other bat (another bat from the same wild roost) and was not the top donor for any other bat. Moreover, BD’s behavior to other adult females changed after Lilith’s death. Before Lilith’s death, BD ranked 3rd and 8th for highest allogrooming and food-sharing rates to other adult females (excluding Lilith). After Lilith’s death, BD’s rank decreased to 9th and 12th for highest allogrooming and food-sharing rates.

These second, third, and fourth paragraphs seem out of order. For example, you talk about Lilith giving birth in the third paragraph after you’ve talked about her decline after giving birth earlier on.

Thanks for pointing this out. We heavily revised this section of text and fixed these issues in structure and order.

55 - 56. “On this day BD was found to be lactating.” I would clarify in the text that she did not likely spontaneously start lactating exactly on the 28th of August. Unfortunately, you cannot rule out that she gradually started beforehand which, makes more sense physiologically. If you can say for certain,

elaborate a bit on how you know or how you 'first noticed her lactating on...' Discuss this caveat and how this relates to when we may have started supplementing the pup with milk. Please note on the next page line 24 through 25 you also mention spontaneously lactating. I would call this something else. For example, you could call it induced lactation or similar. Again, spontaneous would imply no hormonal nor behavioral priming which you have no evidence of (hormonal) and possible behavioral priming given earlier bouts of grooming.

We deleted the phrase "spontaneous" and rewrote some methods and results according to a previous revision (see above).

From the results:

During this time, observations from video footage suggested that BD began to nurse the pup, which seemed to increase gradually.

We also changed the language when referencing previous observations in the discussion:

This female was not lactating before her interactions with the orphaned pup; its previous pup was born two years prior and had died 16 months before the adoption.

Fourth paragraph line 60 to the end of page.

Here you refer to BD not being the top groomer nor food donor. However, it is unclear to me when this statement is for the whole study, before or after she adopts the pup. Please clarify.

We have now addressed this issue (see third revision up from here). Some corresponding text:

Before Lilith's death, BD ranked 3rd and 8th for highest allogrooming and food-sharing rates to other adult females (excluding Lilith). After Lilith's death, BD's rank decreased to 9th and 12th for highest allogrooming and food-sharing rates.

Page 5 of 8 continued.

46 - 49. How often does a single female have more than one pup in a single year? If it is not particularly common, she may benefit more from taking care of a possible help her relative to trying to squeeze in yet another pup in a year. I'm curious which would be the more energetically beneficial strategy.

We now better address this in the discussion. Revised text:

First, unlike most bats, female vampire bats often lack reproductive seasonality and can reproduce year-round, such that a female which adopts an unrelated pup will presumably reduce its own reproductive success, even if it recently lost a pup.

Third, relative to other bats, alloparental care poses an extreme energetic and opportunity cost. Female vampire bats give birth to a single pup after a gestation period of five to seven months [26,39], and a new mother will carry its pup for one to two months, during which time the weight of the pup will at least double. After four to five months, pups will have grown four-fold in mass and will begin to fly and feed on blood [26]. Weaning does not occur until approximately nine months of age [26,40]. In contrast, weaning in other bat species in the same family occurs after only one to three months [41-43]. Finally, reciprocal food-sharing behavior among non-kin adults is likely based on the

co-option of cooperative traits that originated for maternal care [28,31]. The same co-option of traits and expansion of cooperative behavior from kin to non-kin may also help to explain non-kin adoption.

50 - 51. Here is where you discuss the low kinship value. Again, this ought to be mentioned much earlier as it is unclear how you know that these 2 females are unrelated.

We addressed this above when we provided that the capture sites 340 km apart.

Also, this paragraph.

I think you could highlight just how enormously expensive 9 months of lactation would truly be to help illustrate the cost of this unique behavior.

Indeed, female vampire bats put tremendous investment into their young! Thank you for noting that we should highlight this point more clearly. When discussing the timing at which female bats wean their pups, we updated the text (see above).

Page 6 of 8.

2 - 3. Could this occur vice versa where maternal care was co-opted to lead to the origins of food sharing? I suspected this could well be what happened in early humans.

This is actually the idea we meant to convey. That is, maternal care traits (e.g. food sharing to pups) were co-opted for non-kin cooperation (e.g. food sharing between adults). Thank you for pointing out that this was not clear. We have revised accordingly.

Finally, reciprocal food-sharing behavior among non-kin adults is likely based on the co-option of cooperative traits that originated for maternal care [28,31]. The same co-option of traits and expansion of cooperative behavior from kin to non-kin may also help to explain non-kin adoption.

Line 8.

Your discussion really focuses on the social interactions of the 2 adult females but here you discuss interactions with the orphan prior to the mother's death. Can you carefully parse these ideas up and either only focus on one or make sure you discuss the importance of interactions at both levels?

We aim to emphasize how the adopter's behavior changed through time towards both the mother and the young pup, and how her relationship with both bats was unique in comparison to all other bats in the colony. As per the results, the adopter had begun to alloparent the pup as the mother fell ill. At the same time, the adopter increased her rate of food sharing with the mother before the mother died. In line with this evidence, we briefly state in the first paragraph of the discussion that the adopter "responded to changes in need of both the mother and the pup." To add more clarity, we have also revised some text in the discussion on page 6:

Hobaiter et al. [11] suggested that chimpanzees may adopt an unrelated orphan if they previously experienced positive social interactions with that orphan prior to the mother's death. If so, then social interactions between adopter and adopted orphan may also be facilitated by a close social or affiliative relationship between the adopter and the biological parent.

Paragraph lines 15 through 29.

I did not quite follow what was meant by "parental guided matchmaking, sexual imprinting" and to

some extent even “deceit.” If an adoptive mother is actually feeding a pup then she is truly giving the pup some benefit- so are you suggesting that the pup is expected to help later on but ultimately does not? Clarify.

The ‘match-making’ hypothesis suggests that parents adopt unrelated individuals that would ultimately become mates for their own biological offspring. Deceit implies that the adopter “deceives” the adopted individual into recognizing the adopter as kin. As a result, the adopted individual will continue to help at the nest long into the future (given a cooperative breeding system). To clarify these points in the manuscript, we have revised the text:

Some authors have suggested ways in which non-kin adoption might be adaptive for the adopter, including reciprocity [13,45], ‘match-making’ between biological and adopted young to form compatible mating pairs [46], or kinship deceit, by which adopters exploit kin-recognition heuristics and deceive adopted young into a false perception of kinship, thereby causing these young to later help at the nest [47].

Figure 1A timeline.

I would add important key events onto this figure. For example: When did the 2 adults meet? When was Lilith removed? When was Lilith returned? You could also add on hatched bars for the duration of lactation to highlight the period over which this is occurring for each of the 2 mothers. For consistency sake I would use the names in parentheses on this figure as well.

We have altered the language of Figure 1A to clarify. See revised figure above and in attached proof.

Figures 1 B etc.

I would give a number in parenthesis dashed grey lines to highlight the focus possible number of interactions. This gets back to my previous question regarding whether BD isn't just a ‘giver’ and if her behavior changed after she adopted the pup.

We now address this in our revised results section and Figure 2 (see above).

Reviewer: 4

Comments to the Author(s)

I have reviewed Rezik et al’s manuscript titled ‘Non-kin adoption in the common vampire bat’. I thought this was a well-written, succinct manuscript. The article describes a single incidence of offspring adoption in the common vampire bat. This behavior is of interest because it is exceptionally rare, especially among non-kin. Suspected adoption in common vampires was reported in a prior manuscript, but this is the first to make detailed observations that confirm the behavior.

I think the introduction was well written, touching on all appropriate background, and the methods were clear. The use of polynomial regression fitting was a reasonable approach to characterizing the data.

In the results, the timeline was a bit awkward. In the paragraph starting line 41, activities after Lilith’s death were described. Then, the paragraph that starts on line 48 jumps back to describing when Lilith gave birth. To me, this would have better flow if the sentence starting ‘On 9 August 2019..’ was removed and the next started ‘In the weeks following parturition,...’. The date of birth was described

previously in the methods.

I thought the figures nicely portrayed the patterns described.

Thank you for the feedback! Yes, we received a similar comment from another reviewer regarding the structure of the results section and revised this section accordingly.

In the conclusion, the authors state that social interactions strongly impact non-kin adoption. While I agree that is likely, I strongly recommend that authors tone this down, as it is impossible to draw strong conclusions from a single adoption event.

Overall, I think this paper will be of interest to bat biology and the organismal biology community. The biggest problem with this manuscript is that it is based on a single observation of adoption. Thus, the authors and readers should not be confident that the observed behaviors would be typical of adoption in this species, and it remains unclear how common this behavior is. In the past, journals such as the Journal of Mammalogy had a 'Notes' section for important natural history observations, such as this one. Few journals will publish this type of natural history observation based on a single observation – it's up to the editors if natural history observations have a home in the Royal Society Open Science. I think natural history observations are important, and it would be supportive of Open Science publishing this manuscript if it meets their standards.

We completely agree. It is hard to gauge how common this behavior might be, especially in wild populations. Vampire bat adoption has only been rarely observed, and only in captivity. As a result, we have added more conservative language throughout the discussion and conclusion, urging caution in the interpretation of our observations. We also strongly agree with the reviewer that observations of this nature are of wide interest, and there is great value in making them available to the scientific community.